# Open-Set Learning for Addressing Label Skews in One-Shot Federated Learning

## Abstract

Federated learning (FL) is crucial for collaborative model training, yet it faces significant challenges from data heterogeneity, particularly label skews across clients, where some classes may be underrepresented or absent entirely. In one-shot FL, where clients only communicate with the server once, this problem becomes even more challenging. Recent solutions propose incorporating open-set learning (OSL) to tackle this issue by detecting unknown samples during inference, but current methods like FedOV lack adaptability to varying client data distributions. In this paper, we provide a theoretical analysis proving that improving OSL algorithms can effectively address label skews in one-shot FL, since one-shot FL is learnable through good OSL algorithms regardless of label skews. We also empirically evaluate state-of-the-art OSL algorithms and identify their limitations. Based on these insights, we propose FedAdav, an adaptive algorithm that combines OSL signals to significantly improve ensemble accuracy in one-shot FL under label skews. Through extensive experiments, we demonstrate that exploring better OSL is key to overcoming label skew challenges in federated learning.

## 1 Introduction

Federated learning (FL) (McMahan et al., 2016) enables collaborative model training across distributed clients without the need to share raw data. While traditional FL requires multiple communication rounds to achieve strong model performance, one-shot FL (Guha et al., 2019) reduces this overhead by limiting communication to a single round. This efficiency makes one-shot FL particularly useful for applications such as model hubs (Vartak, 2016), where pre-trained models are shared and aggregated to form powerful ensembles across multiple data sources (Wang et al., 2023).

A major challenge in one-shot FL is label skew (Li et al., 2021b), where clients have highly imbalanced or non-overlapping class distributions. In practice, this occurs when some clients have abundant data for certain classes, while other clients have limited or no data for those classes. For example, in healthcare, disease prevalence varies significantly across regions, causing models trained on localized data to be biased towards certain classes. Label skew can severely affect the performance of one-shot FL, as the lack of iterative communication rounds prevents the model from adjusting to these imbalances. Without addressing this issue, the global model may fail to generalize well across the combined data from clients.

To address the problem of label skew, recent research has applied open-set learning (OSL) to one-shot FL. OSL allows models to classify known classes while detecting and rejecting out-of-distribution (OOD) samples that the model hasn't encountered during training (Salehi et al., 2021; Ye et al., 2022). This capability is especially valuable in applications like autonomous driving and healthcare, where unknown situations can arise. For instance, FedOV (Diao et al., 2023) introduces an "unknown" class by generating outliers from corrupted training data to handle unseen classes. While effective in addressing label skew, FedOV only relies on a single OSL signal from corrupted data samples, limiting its flexibility.

In this paper, we tackle the challenges posed by label skews in one-shot FL by exploring both theoretical and empirical perspectives. Our theoretical analysis proves the learnability of one-shot FL with OSL, emphasizing the importance of improved OSL algorithms for handling label skews. Based on these insights, we propose FedAdav, a novel adaptive algorithm that combines multiple OSL signals to enhance one-shot federated ensemble accuracy. Through comprehensive experiments,

we demonstrate that FedAdav outperforms state-of-the-art (SOTA) OSL algorithms under severe label skew conditions.

Our contributions are summarized as follows:

- **Theoretical Contribution:** We prove the learnability of one-shot FL ensembles with OSL. This is the first theoretical study of one-shot federated ensembles using OSL, underscoring the role of OSL in mitigating label skews in one-shot FL.
- **Algorithmic Improvement:** Based on our theoretical analysis, we propose an OSL algorithm, FedAdav, that combines the strengths of multiple OSL methods, significantly improving performance under label skews.
- **Empirical Evaluation:** Extensive experiments on FedAdav and 11 other state-of-the-art OSL algorithms demonstrate that FedAdav achieves superior accuracy under severe label skew conditions, corroborating our theoretical results.

The remainder of the paper is organized as follows: Section 2 reviews related works, Section 3 presents our theoretical analysis, Section 4 describes our proposed method, and Section 5 discusses the experimental results.

## 2 RELATED WORKS

### 2.1 ONE-SHOT FL AND LABEL SKEWS

Federated learning (FL) enables training machine learning models across multiple clients without requiring raw data to be shared (McMahan et al., 2016; Kairouz et al., 2019; Yang et al., 2019; Li et al., 2019). In one-shot FL, clients communicate with the server only once, significantly reducing communication costs compared to traditional multi-round FL (Guha et al., 2019). Applications such as model hubs benefit from one-shot FL by allowing pre-trained models to be aggregated from multiple data sources, creating a powerful ensemble (Vartak, 2016; Wang et al., 2023).

However, label skews, where clients have very imbalanced or non-overlapping class distributions, pose a major challenge in one-shot FL. As each client trains on locally skewed data, the global model is biased towards the dominant classes in each client, leading to poor generalization. This is particularly problematic for one-shot FL, since the lack of multiple communication rounds prevents the model from correcting imbalances or filling gaps in class distributions (Li et al., 2021b).

To address the label skews in one-shot FL, FedOV (Diao et al., 2023) proposes to create an "unknown" class by generating corrupted versions of in-distribution data during local training, which are then treated as outliers. Clean data and corrupted data are put into a batch for training. The model learns to distinguish known classes from outliers, enhancing its ability to handle label skews in one-shot FL. The loss function for FedOV is:

$$L_{\text{FedOV}} = -\sum_{i=1}^{K} y_i \log p_i - y_u \log p_u, \tag{1}$$

where $y_i$ is the true label for the known classes, $p_i$ is the predicted probability for class $i$, $y_u$ is the label for the "unknown" class, and $p_u$ is the predicted probability for the "unknown" class. FedOV is effective in improving one-shot FL accuracy by introducing the "unknown" class, but its reliance on generated outliers as a primary OOD detection signal limits its OOD detection capability, especially under severe label skews.

### 2.2 OPEN-SET LEARNING (OSL)

Traditional closed-set learning focuses on classifying known classes, but open-set learning (OSL) extends this by enabling models to detect and reject out-of-distribution (OOD) samples (Salehi et al., 2021; Ye et al., 2022). OSL is essential in safety-critical applications, such as autonomous driving, healthcare, and cybersecurity, where unknown or unseen data can emerge during deployment. For instance, a self-driving car may encounter objects or scenarios that are not part of the training set, and failing to correctly identify them as unknown can lead to dangerous consequences.

Among popular OSL methods, maximum softmax probability (MSP) (Hendrycks & Gimpel, 2017) and maximum logit score (MLS) (Hendrycks et al., 2022a) are often used for scoring outliers. However, these approaches primarily focus on OOD detection based on a single score function, without leveraging additional signals or tasks.

RotPred (Hendrycks et al., 2019b) is a self-supervised OSL method that adds a pretext task to help the model learn better feature representations. Specifically, RotPred requires the model to predict the rotation of input images (0°, 90°, 180°, or 270°). This task forces the model to extract more meaningful features to improve its ability to detect outliers, especially in cases where the training data is limited or imbalanced. The loss for RotPred is given by:

$$L_{\text{RotPred}} = -\sum_{r=1}^{4} y_r \log p_r, \tag{2}$$

where $y_r$ is the true rotation label, and $p_r$ is the predicted probability for the corresponding rotation. Despite its effectiveness, RotPred can struggle on datasets where the rotations cannot be differentiated from the original data, such as in the case of digits 0 and 1.

Existing OSL benchmarks primarily evaluate methods on static tabular and image datasets. For example, KNN-based outlier detection is tested on tabular data (Campos et al., 2016), while both shallow and deep anomaly detection methods are benchmarked in works like (Ruff et al., 2021). Recent efforts like ADBench (Han et al., 2022) and OpenOOD (Yang et al., 2022) test OSL and OOD detection on datasets such as MNIST, CIFAR-10, and ImageNet.

However, these benchmarks focus on score functions and metrics like AUROC and FPR@95, which are effective in traditional settings but may not suit one-shot FL with label skews. In one-shot FL, where client data distributions are imbalanced, these standard metrics can struggle. Setting thresholds for OOD detection becomes critical, yet existing methods such as extreme value theory (EVT) (Coles et al., 2001) and percentile-based thresholds (Liang et al., 2017) are under-explored in federated settings. Thus, prior OSL benchmarks, while useful, may not fully capture the complexities of OSL in one-shot FL with label skews.

### 2.3 RELATED THEORETICAL STUDIES

Prior FL theoretical works usually prove the convergence of specific algorithm (Wang et al., 2021; Mohri et al., 2019), personalized FL (Deng et al., 2020; Hanzely et al., 2020) or incentive mechanisms (Kang et al., 2019). One-shot FL theory is under-explored. We only find one recent theoretical work (Jhunjhunwala et al., 2023) on one-shot FL. However, its focus is on Fisher Information and the convergence proof is restricted to sufficiently wide two-layer ReLU network. Our proof focuses on one-shot FL open-set ensemble and does not depend on network structure. Another related work (Zhu et al., 2023) studies multi-round FL on OSL tasks of medical applications, while ours focuses on applying OSL to address label skews in one-shot FL.

Mostly related to our work, the theoretical studies on class-incremental learning (CIL) (Kim et al., 2022; 2023) show that OSL and CIL can improve each other, and CIL can be learned by OSL. CIL is similar to one-shot FL in that the training set is divided into several tasks with different classes, and the raw data of past tasks (except a few exemplars) are not accessible. However, there are three main differences when applying them to one-shot FL. (1) CIL assumes that the classes of each task do not overlap, while FL clients can have overlapping classes. (2) CIL learns from a sequence of tasks and can exploit information from stored exemplars or previous models, while in one-shot FL, the client cannot access any data or model of other clients. (3) The form of one-shot FL ensemble is very different from the final model in CIL.

## 3 THEORETICAL ANALYSIS

In this section, we first prove that a good OSL algorithm naturally leads to a good one-shot FL ensemble. Then, we prove the learnability of one-shot FL ensemble with OSL. Theoretically, the task of one-shot FL ensemble can be addressed by exploring good OSL algorithms.

The notations for theoretical analysis are listed in Appendix B. Suppose there are $N$ clients. Denote the dataset as $\mathcal{D}$, the feature space as $\mathcal{X}$, the label space as $\mathcal{Y}$, and the hypothesis space as $\mathcal{H}$. The

number of labels is $K$, i.e. $|\mathcal{Y}| = K$. Denote $Y^i \subset \mathcal{Y}$ as the set of classes that can be seen in client $i$. Naturally $\bigcup_{i=1}^{N} Y^i = \mathcal{Y}$. Each client trains an OSL model and the server uses their ensemble as the final model.

For client $i$, we can decouple its OSL model into (1) an OOD detection model $u^i : \mathcal{X} \to [0, 1]$, which measures the known confidence of the input; and (2) a closed-set classification model $f_c^i : \mathcal{X} \to [0, 1]^K$, which outputs a probability distribution that the input belongs to each known class of client $i$. Additionally, if class $y$ is unseen in client $i$, i.e. $y \notin Y^i$, we assume $f_c^i(x)_y = 0$ since the closed-set classifier is supposed to assign probabilities only among seen classes.

Given $N$ OSL models, we have the global ensemble

$$F(x) = \sum_{i=1}^{N} u^i(x) f_c^i(x). \tag{3}$$

Therefore, we have the empirical loss of the ensemble

$$\mathcal{L}(F(x); y) = -\log \frac{\sum_{i=1}^{N} u^i(x) f_c^i(x)_y}{\sum_{i=1}^{N} u^i(x)}. \tag{4}$$

Further, we denote the closed-set classification loss of client $i$ as

$$\mathcal{L}(f_c^i(x); y) = -\log f_c^i(x)_y, \tag{5}$$

and the OOD detection loss as

$$\mathcal{L}(u(x); y) = -\log \frac{\sum_{i=1}^{N} \mathbf{1}_{y \in Y^i} u^i(x)}{\sum_{i=1}^{N} u^i(x)}. \tag{6}$$

The closed-set classification loss is just a cross-entropy loss. For the part of OOD detection loss, our task becomes selecting a correct client who actually sees $x$. Different from the assumption in (Kim et al., 2022), in the FL setting we regard any client with class $y$ as the correct target. Therefore, the probability of correctly choosing the client with class $y$ is exactly $\frac{\sum_{i=1}^{N} \mathbf{1}_{y \in Y^i} u^i(x)}{\sum_{i=1}^{N} u^i(x)}$. Like cross-entropy loss, we take the negative logarithm of such probability as the OOD detection loss.

The relationship between OSL (closed-set classifier plus OOD detector) and CIL is shown in (Kim et al., 2022). Inspired by it, we can theoretically prove that the one-shot FL ensemble benefits from better OSL algorithms.

**Theorem 3.1.** $\forall (x, y) \sim \mathcal{D}$, if $\mathbf{1}_{y \in Y^i} \mathcal{L}(f_c^i(x); y) \leq \varepsilon$ and $\mathcal{L}(u(x); y) \leq \delta$, $\forall 1 \leq i \leq N$, then $\mathcal{L}(F(x); y) \leq \varepsilon + \delta$.

**Proof Sketch.** The federated ensemble model is formed by aggregating each client's prediction, weighted by their OOD detection score $u^i(x)$. We apply Jensen's inequality to bound this federated ensemble loss. This results in the ensemble loss being no greater than the weighted average of the individual client losses.

The complete proof is given in Appendix B.2.

**Implication.** If we have a better OSL algorithm with lower closed-set classification loss and lower OOD detection loss, the one-shot FL ensemble will also have lower loss. Moreover, an ideal OSL algorithm can be a perfect solution for FL ensemble. Closed-set ensemble assigns fixed weights to all models and receives misleading votes from overconfident clients. The advantage of open-set voting compared with closed-set voting is better weights $u^i(x)$, which requires proper "unknown" probabilities across the clients.

Recently, (Kim et al., 2023) further gives a proof that CIL is learnable. Inspired by its framework, we further establish the learnability theory for one-shot federated learning.

Our analysis is based on the learnability of OOD detection, which is proved in (Fang et al., 2022). We first consider the case of closed-set learning to demonstrate the necessity of open-set learning. Following the notations in (Kim et al., 2023), we denote $D_i$ as the data distribution on client $i$. We

then define a weighted distribution

$$D_{[1:N]} = \sum_{i=1}^{N} \pi_i D_i \Big/ \sum_{j=1}^{N} \pi_j, \tag{7}$$

where weights $\{\pi_i\}_{i=1}^{N}$ are positive real values. The summation notation here indicates a new distribution, i.e., to sample $x$ from a client dataset $D_i$ under probability $\pi_i \big/ \sum_{j=1}^{N} \pi_j$. Besides, denote the risk function as

$$\mathbf{R}_D(F) = \mathbb{E}_{(x,y) \sim D}[\mathcal{L}(F(x); y)]. \tag{8}$$

It is the expectation of the loss of a global ensemble function $F$ on a dataset $D$.

**Definition 3.2** (Fully-Observable Closed-Set Learnability). Given $\mathcal{D}$ and $\mathcal{H}$, a FL framework is learnable if there exist closed-set classifiers $\{f_c^i\}_{i=1}^{N}$, a closed-set global ensemble $F(x) = \sum_{i=1}^{N} f_c^i(x)/N$, and a sequence $\{\varepsilon_n | \lim_{n \to \infty} \varepsilon_n = 0\}$, such that $\forall D_1, \cdots, D_N \in \mathcal{D}$ and $\forall \pi_1, \cdots, \pi_N > 0$:

$$\mathbb{E}_{S \sim D_{[1:N]}} \left[ \mathbf{R}_{D_{[1:N]}}(F) - \inf_{G \in \mathcal{H}} \mathbf{R}_{D_{[1:N]}}(G) \right] < \varepsilon_n. \tag{9}$$

**Definition 3.3** (Partially-Observable Closed-Set Learnability). Given $\mathcal{D}$ and $\mathcal{H}$, a FL framework is learnable if there exist $\{f_c^i\}_{i=1}^{N}$ and a sequence $\{\varepsilon_n | \lim_{n \to \infty} \varepsilon_n = 0\}$, such that $\forall D_1, \cdots, D_N \in \mathcal{D}$:

$$\max_{i=1, \cdots, N} \mathbb{E}_{S \sim D_i} \left[ \mathbf{R}_{D_i}(f_c^i) - \inf_{g_i \in \mathcal{H}} \mathbf{R}_{D_i}(g_i) \right] < \varepsilon_n. \tag{10}$$

In Definition 3.2, the loss function of closed-set ensemble is

$$\mathcal{L}(F(x); y) = -\log \left( \sum_{i=1}^{N} f_c^i(x)_y / N \right). \tag{11}$$

In Definition 3.3, the loss function of each closed-set classifier is

$$\mathcal{L}(f_c^i(x); y) = -\log f_c^i(x)_y. \tag{12}$$

It would be desirable if Definition 3.3 can imply Definition 3.2. But we can show that without OOD detection, such transformation cannot hold:

**Theorem 3.4** (Definition 3.3 does not imply Definition 3.2). *Assume $\mathcal{H}$ is big enough so that $\inf_{G \in \mathcal{H}} \mathbf{R}_{D_{[1:N]}}(G) = 0$ and the measure of $\mathcal{D}$ is finite, i.e. $m(\mathcal{D}) < \infty$. If there exist classes unseen by some clients, i.e. $r = m(\{(x,y) : y \text{ is not seen in all } Y^i\})/m(\mathcal{D}) > 0$, then there exist $\{f_c^i\}_{i=1}^{N}$, and $F(x) = \sum_{i=1}^{N} f_c^i(x)/N$ trained by some sample set $S$ trained by some sample set $S$, satisfying Definition 3.3 but not Definition 3.2.*

**Proof Sketch.** Since the closed-set global ensemble is $F(x) = \sum_{i=1}^{N} f_c^i(x)/N$, if a client has not seen a class, it will still contribute a non-zero classification probability for other classes, leading to errors in the ensemble output. Without OOD detection $u^i(x)$, these contributions from clients lacking certain class data bias the ensemble, causing errors for classes they have not observed. Hence, closed-set learning cannot address label skews effectively.

The complete proof is given in Appendix B.3.

**Implication.** Without OOD coefficients, the closed-set ensemble has to sum up the undesirable clients' results, which inevitably leads to some errors under label skews.

To address such limitation, we need OOD detection function $u^i : \mathcal{X} \to [0, 1]$. Similar with the notation from Kim et al. (2023), let $O_i$ be the OOD set of client $i$, $D_i^{\alpha_i} = (1 - \alpha_i)D_i + \alpha_i O_i, \alpha_i \in [0, 1)$, and $D_{[1:N]}^{\alpha_{[1:N]}} = \sum_{i=1}^{N} \pi_i D^{\alpha_i} \big/ \sum_{i=1}^{N} \pi_i$. Correspondingly, closed-set classifier $f_c^i : \mathcal{X} \to [0, 1]^K$ should be expanded to open-set classifier $f_o^i : \mathcal{X} \to [0, 1]^{K+1}$, where

$$f_o^i(x)_y = \begin{cases} u^i(x) f_c^i(x)_y, & y = 1, ..., K, \\ 1 - u^i(x), & y = K + 1. \end{cases} \tag{13}$$

The sum of all elements of $f_o^i$ is still 1, therefore $\{f_o^i\}$ is still a distribution. Then we have the following definitions.

**Definition 3.5** (Fully-Observable Open-Set Learnability). Given $\mathcal{D}$ and $\mathcal{H}$, a FL framework is learnable if there exist $\{f_c^i\}_{i=1}^N$, $\{u^i\}_{i=1}^N$, an open-set global ensemble $F(x) = \frac{\sum_{i=1}^N u^i(x) f_c^i(x)}{\sum_{i=1}^N u^i(x)}$ and a sequence $\{\varepsilon_n | \lim_{n\to\infty} \varepsilon_n = 0\}$, such that $\forall D_1, \cdots, D_N \in \mathcal{D}, \forall \pi_1, \cdots, \pi_N > 0, \forall O_1, \cdots, O_N \in \mathcal{D}$, and $\forall \alpha_1, \cdots, \alpha_N \in [0, 1)$:

$$\mathbb{E}_{S \sim D_{[1:N]}} \left[ \mathbf{R}_{D_{[1:N]}^{\alpha_{[1:N]}}}(F) - \inf_{G \in \mathcal{H}} \mathbf{R}_{D_{[1:N]}^{\alpha_{[1:N]}}}(G) \right] < \varepsilon_n. \tag{14}$$

**Definition 3.6** (Partially-Observable Open-Set Learnability). Given $\mathcal{D}$ and $\mathcal{H}$, a FL framework is learnable if there exist $\{f_o^i\}_{i=1}^N$ from Equation 13, and a sequence $\{\varepsilon_n | \lim_{n\to\infty} \varepsilon_n = 0\}$, such that $\forall D_1, \cdots, D_N \in \mathcal{D}, \forall O_1, \cdots, O_N \in \mathcal{D}$, and $\forall \alpha_1, \cdots, \alpha_N \in [0, 1)$:

$$\max_{i=1,\cdots,N} \mathbb{E}_{S \sim D_i} \left[ \mathbf{R}_{D_i^{\alpha_i}}(f_o^i) - \inf_{g_i \in \mathcal{H}} \mathbf{R}_{D_i^{\alpha_i}}(g_i) \right] < \varepsilon_n. \tag{15}$$

For open-set ensemble, we have $\mathcal{L}(F(x); y) = -\log \frac{\sum_{i=1}^N u^i(x) f_c^i(x)_y}{\sum_{i=1}^N u^i(x)}$. We assume that in FL settings, all classes in the test set are observed by at least one client. Therefore we disregard the "unknown" dimension of the open-set ensemble. However, when testing the local open-set classifier, the loss of unseen classes should be calculated by the "unknown" dimension, and we have

$$\mathcal{L}(f_o^i(x); y) = -\log f_o^i(x)_y \tag{16}$$
$$= -\mathbf{1}_{y \in Y^i} \log u^i(x) f_c^i(x)_y - \mathbf{1}_{y \notin Y^i} \log(1 - u^i(x)). \tag{17}$$

**Theorem 3.7** (Definition 3.6 implies Definition 3.5). *Assume $\mathcal{H}$ is big enough so that $\inf_{G \in \mathcal{H}} \mathbf{R}_{D_{[1:N]}^{\alpha_{[1:N]}}}(G) = 0$. If there exist $\{f_c^i\}_{i=1}^N$, $\{u^i\}_{i=1}^N$, and $\{f_o^i\}_{i=1}^N$ according to Equation 13 satisfying Definition 3.6, then $\{f_c^i\}_{i=1}^N$, $\{u^i\}_{i=1}^N$ satisfy Definition 3.5.*

**Proof Sketch.** The open-set federated ensemble loss is split into two parts: 1) closed-set classification loss, which measures the accuracy on known (in-distribution) samples; 2) OOD detection loss, which measures the effectiveness of identifying out-of-distribution samples. This setup mitigates the effects of label skews, as clients not confident about certain classes do not introduce errors into the ensemble. Based on the learnability of OOD detection in each client, we can prove the learnability of open-set federated ensemble.

The complete proof is given in Appendix B.4.

**Implication.** This theorem shows that with OOD detection, one-shot federated learning learnability can be ensured. Combined with the proof of OOD detection learnability in (Fang et al., 2022), we can establish the learnability of one-shot FL by open-set voting. Thus, exploring good OSL algorithms with both low closed-set classification loss and low OOD detection loss is the key to addressing the label skews in one-shot FL.

## 4 METHODS: ADAPTIVE OSL TO ADDRESS LABEL SKEWS IN ONE-SHOT FL

Our theoretical findings reveal that improving open-set learning (OSL) is key to enhancing performance in one-shot FL, particularly when dealing with label skews. To address this, we propose FedAdav, a novel adaptive algorithm that integrates multiple OSL methods to overcome the limitations of existing approaches.

In this section, we first discuss two state-of-the-art OSL methods, FedOV and RotPred, which serve as the foundation for FedAdav. Then, we describe how FedAdav adaptively combines these approaches to effectively handle the challenges posed by label skews in one-shot FL.

### 4.1 FEDOV AND ROTPRED: BASELINE OSL METHODS

**FedOV** (Diao et al., 2023) is SOTA algorithm to tackle label skews in one-shot FL by incorporating OSL. It works by generating outliers from corrupted in-distribution samples, which are then assigned to an "unknown" class. This allows the model to distinguish between known and unknown classes, providing a way to handle the missing classes in the client datasets due to label skews. However,

FedOV relies heavily on these generated outliers as the sole signal for OOD detection, limiting its adaptability in diverse real-world scenarios where outliers may exhibit varied characteristics.

**RotPred** (Hendrycks et al., 2019b) is another popular OSL method that improves outlier detection by adding a self-supervised task—predicting the rotation of input images. By forcing the model to predict one of four rotation angles (0°, 90°, 180°, or 270°), RotPred enhances the feature learning process, which aids in OOD detection. While RotPred has proven effective in certain contexts, it can struggle with datasets where the notion of rotation is ambiguous or uninformative.

Both methods have demonstrated strong results individually, but their limitation motivate the need for a more flexible and adaptive approach.

## 4.2 FEDADAV: FEDERATED ADAPTIVE VOTING

FedAdav builds on the strengths of both FedOV and RotPred by adaptively combining their respective signals. The goal is to enhance outlier detection and classification accuracy in one-shot FL ensembles by dynamically adjusting the use of these signals during training and inference.

**Training Procedure.** Each client in FedAdav initially trains a local model using the sum of FedOV and RotPred loss functions. The combined loss is $L = L_{\text{FedOV}} + \lambda L_{\text{RotPred}}$. $L_{\text{FedOV}}$ is the outlier detection loss from FedOV, which classifies in-distribution samples and corrupted outliers. $L_{\text{RotPred}}$ is the self-supervised loss from RotPred, which predicts the rotation of input images. $\lambda$ is the trade-off parameter between two losses.

The total loss is adaptively adjusted during training. After a given number of rounds $T_{\text{check}}$, if the rotation prediction loss $L_{\text{RotPred}}$ still exceeds a predefined threshold $\tau$, FedAdav disables the RotPred component and continues training solely with FedOV. In such case, we assume that the rotation prediction loss is not a good indicator for the train dataset. Since rotation prediction is a 4-way classification task, we set $\tau = 0.5$. According to empirical results, this threshold can filter out classes that cannot be predicted rotations, e.g., digit 0, 1, or classes containing different angles. This adaptive training mechanism ensures that the model only uses the most relevant signals, avoiding the computational overhead of training on tasks that do not contribute to improved performance.

**Inference Procedure.** During inference, FedAdav uses a combination of FedOV and RotPred signals to make predictions. If the rotation prediction loss is high (indicating high uncertainty in predicting the rotation), the sample is rejected as an outlier. Otherwise, the sample is processed by FedOV, and the predictions are summed into the final decision.

The whole procedure is shown in Algorithm 1.

---

**Algorithm 1:** The FedAdav algorithm. $\sigma$ is the softmax function.

**Input:** number of clients $N$, number of classes $c$, training rounds $T$

1 **Each client executes**:
2 Initialize local model $f_i$
3 useRotPred$_i$ = True
4 **for** $t = 1, ..., T$ **do**
5     **if** useRotPred$_i$ **then**
6        $L = L_{\text{FedOV}} + \lambda L_{\text{RotPred}}$
7     **else**
8        $L = L_{\text{FedOV}}$
9     Update $f_i$ with loss $L$
10     **if** $t == T_{\text{check}}$ and $L_{RotPred} > \tau$ **then**
11        useRotPred$_i$=False

12 **if** useRotPred$_i$ **then**
13     Calculate RotPredBar$_i$
14 Upload $f_i$, useRotPred$_i$, RotPredBar$_i$ to server

15 **Server executes**:
16 Collect uploaded information from clients.

17 **Prediction**($x$):
18 scores= $\mathbf{0}$
19 **for** $i = 1, ..., N$ **do**
20     **if** useRotPred$_i$ and
21     $L_{\text{RotPred}} > $ RotPredBar$_i$ **then**
22        Pass // `Ignore samples whose rotations cannot be predicted`
23     **else**
24        scores=scores+$\sigma(f_i(x))$
25 $y_p = \arg\max_{j \in \{0,1,2,...,c-1\}} \text{scores}_j$
26 **return** $y_p$

---

## 5 EXPERIMENTS

### 5.1 SETUPS

**Datasets.** We experiment with three image classification datasets with increasing difficulty level: MNIST LeCun et al. (1998), CIFAR-10 (Krizhevsky et al., 2009), and CIFAR-100 (Krizhevsky et al., 2009). By default, we set 10 clients for MNIST and CIFAR-10, and 100 clients for CIFAR-100 to test scalability. We adopt the FL label skew partitioning strategy in NIID-Bench (Li et al., 2021b). $\#C = k$ assigns $k$ unique labels for each client, while $p_k \sim Dir(\beta)$ divides each class samples using Dirichlet distribution.

**Models.** We experiment with two neural network architectures. In accordance with OpenOOD (Yang et al., 2022), we use the LeNet for MNIST and ResNet-18 for CIFAR-10 and CIFAR-100.

**Baselines.** Besides the SOTA one-shot FL algorithm FedOV (Diao et al., 2023) and naive closed-set ensemble baseline, we select a total of 9 SOTA OSL algorithms from OpenOOD (Yang et al., 2022) to apply to the one-shot FL setting. Specifically, we select the top 10 OSL algorithms from CIFAR-10 and CIFAR-100 leaderboards, and remove those requiring external outlier examples or local data embedding information to suit the FL settings.

**Hyper-parameters.** By default, we use Adam optimizer with learning rate 0.001, and train 200 local epochs with batch size 64. For FedAdav, we set $\lambda = 1, T_{\text{check}} = 50$ and $\tau = 0.5$ by default.

Finally our comparison includes 11 baselines: 1) FedOV (Diao et al., 2023), 2) CE+MSP (naive closed-set ensemble baseline), 3) RotPred (Hendrycks et al., 2019b), 4) PixMix (Hendrycks et al., 2022b)+MSP, 5) RotPred+PixMix, 6) LogitNorm (Wei et al., 2022)+MSP, 7) Logit-Norm+PixMix+MSP, 8) RegMixup (Pinto et al., 2022)+MSP, 9) CE+GEN (Liu et al., 2023), 10) CE+MLS (Hendrycks et al., 2022a), and 11) CE+ReAct (Sun et al., 2021). CE means normal training with cross entropy loss. MSP (Hendrycks & Gimpel, 2017) is a naive baseline where the maximum softmax probability is regarded as the confidence. Further details are listed in Appendix C.1.

### 5.2 AN OVERALL COMPARISON

We evaluate the performance of FedAdav by benchmarking it against six strong baselines and closed-set training baseline in the one-shot federated learning (FL) setting under severe label skew conditions. The results of the other four relatively weak baselines are detailed in Appendix C.2. According to Table 1, FedAdav achieves the superior accuracy compared to these baselines under severe label skews, reflecting its ability to adaptively combine generated outliers and self-supervised tasks to address label skews in open-set federated ensembles.

Table 1: Test accuracy of OSL algorithms on one-shot FL label skews. We apply 95% TPR threshold to algorithms in OpenOOD for voting according to the convention. All experiments are repeated with three different seeds. In each line, the highest accuracy is bold and the second highest is underlined.

| Dataset | Partition | FedAdav | FedOV | CE+MSP (closed-set) | RotPred +PixMix | LogitNorm+ PixMix+MSP | PixMix +MSP | RotPred | LogitNorm +MSP |
|---|---|---|---|---|---|---|---|---|---|
| MNIST | $\#C = 1$ | **70.9%**$_{\pm 0.8\%}$ | 68.2%$_{\pm 1.9\%}$ | 18.3%$_{\pm 0.4\%}$ | 34.5%$_{\pm 0.4\%}$ | 13.2%$_{\pm 0.3\%}$ | 18.3%$_{\pm 0.4\%}$ | 34.5%$_{\pm 0.5\%}$ | 13.2%$_{\pm 0.3\%}$ |
| | $\#C = 2$ | **78.3%**$_{\pm 10.5\%}$ | 60.0%$_{\pm 7.8\%}$ | 60.1%$_{\pm 3.7\%}$ | 68.1%$_{\pm 2.4\%}$ | 54.4%$_{\pm 0.8\%}$ | 60.1%$_{\pm 3.7\%}$ | 68.2%$_{\pm 1.9\%}$ | 54.1%$_{\pm 0.5\%}$ |
| CIFAR-10 | $\#C = 1$ | **51.7%**$_{\pm 0.7\%}$ | 26.2%$_{\pm 4.1\%}$ | 8.4%$_{\pm 0.1\%}$ | 47.5%$_{\pm 1.3\%}$ | 7.5%$_{\pm 0.4\%}$ | 8.5%$_{\pm 0.1\%}$ | 47.1%$_{\pm 0.7\%}$ | 7.6%$_{\pm 0.4\%}$ |
| | $\#C = 2$ | **58.4%**$_{\pm 5.5\%}$ | 45.4%$_{\pm 2.5\%}$ | 45.4%$_{\pm 1.1\%}$ | 56.2%$_{\pm 2.3\%}$ | 41.4%$_{\pm 3.3\%}$ | 45.2%$_{\pm 1.0\%}$ | 56.3%$_{\pm 2.1\%}$ | 41.2%$_{\pm 3.5\%}$ |
| CIFAR-100 | $\#C = 1$ | **13.9%**$_{\pm 0.6\%}$ | 8.9%$_{\pm 0.6\%}$ | 0.5%$_{\pm 0.0\%}$ | 3.8%$_{\pm 0.1\%}$ | 0.4%$_{\pm 0.1\%}$ | 0.5%$_{\pm 0.0\%}$ | 3.8%$_{\pm 0.0\%}$ | 0.4%$_{\pm 0.0\%}$ |
| | $\#C = 2$ | 15.1%$_{\pm 1.5\%}$ | **15.7%**$_{\pm 0.5\%}$ | 9.2%$_{\pm 0.4\%}$ | 11.0%$_{\pm 0.8\%}$ | 8.9%$_{\pm 0.7\%}$ | 9.3%$_{\pm 0.5\%}$ | 10.0%$_{\pm 0.1\%}$ | 9.0%$_{\pm 0.8\%}$ |

From extensive experiments, we make the following findings.

**In terms of overall accuracy.** (1) To address label skews, it will be good to combine generated outliers and self-supervised learning. FedAdav achieves the best accuracy in 5 out of the 6 settings, and achieves the top 2 accuracy in all settings. FedAdav significantly improves the federated ensemble accuracy compared to the other SOTA OSL algorithms. (2) Pure data augmentation is not suitable to addressing label skews, although they can improve generalization in closed-set learning. PixMix and RegMixup can be viewed as data augmentation methods. Unlike FedOV which regards the destructed data as class "unknown", PixMix and RegMixup view the augmented data as the seen class, which do not directly address the OSL task. (3) Pure post-processing methods (e.g., GEN, MLS

and ReAct) and LogitNorm are not suitable to addressing label skew settings in FL. Under scenarios where the number of classes is limited, the model tends to be over-confident. Especially when there is only one seen class, the closed-set model consistently outputs that class without capturing any useful features, according to information bottleneck theory (Tishby & Zaslavsky, 2015). In such cases, the performance of these methods can be bad, since the closed-set model itself forgets much feature-related information.

**In terms of partition.** In extreme label skew partitions, except FedAdav, FedOV and RotPred, other explored OSL algorithms cannot perform well under such settings. In fact, many prior OSL algorithms (Neal et al., 2018; Yoshihashi et al., 2019b; Perera et al., 2020b; Zhou et al., 2021) are not designed and tested on datasets with one or two seen classes, thus the settings are under explored.

In summary, some prior OSL algorithms cannot be well adapted to one-shot FL settings, especially when the number of classes is limited. PixMix provides marginal improvement ($< 1\%$) over RotPred, LogitNorm and pure closed-set training. Algorithms trained with cross-entropy (CE) loss provide marginal improvement over closed-set ensemble baseline. Thus, our proposed FedAdav combines the advantages of prior algorithms, and finally achieves the SOTA federated ensemble accuracy.

### 5.3 COMPARING THRESHOLDS

In this section, we evaluate the impact of different thresholds by varying the true positive rate (TPR) among $\{0.4, 0.5, 0.6, 0.7, 0.8, 0.9, 0.95, 0.98\}$. Previous works primarily focus on 95% TPR, but we aim to investigate a broader range of thresholds, including TPR values from 0.4 to 0.98, to understand their effect on performance, specifically measuring one-shot federated ensemble accuracy across these thresholds.

Since our overall comparison identified FedAdav, FedOV, and RotPred as the top-performing algorithms, we focus on these methods. Figure 1 shows the results for CIFAR-10. FedAdav and FedOV consistently perform well when the TPR is set between 80-95%, while RotPred shows its best performance around the 80-90% range. This suggests that tuning the TPR within this range can optimize model performance, as evidenced by the varying accuracy results across different setups. More results and discussions are shown in Appendix C.3.

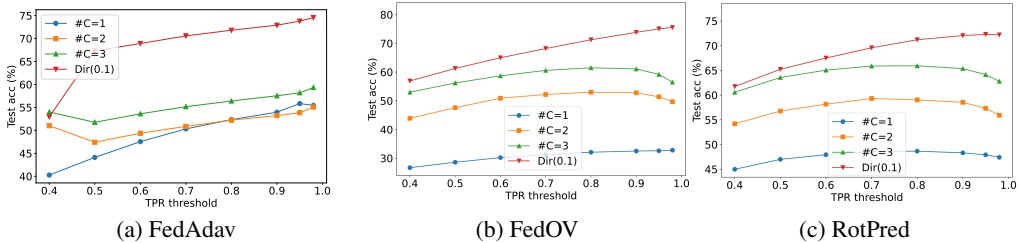

(a) FedAdav          (b) FedOV          (c) RotPred

Figure 1: Ensemble accuracy on CIFAR-10 using different TPR threshold values.

### 6 CONCLUSION

This paper explores the integration of open-set learning (OSL) into one-shot federated learning (FL) ensembles, with a particular focus on addressing the challenge of label skews. We provide both theoretical insights and empirical validation. Our theory proves that one-shot FL can be effectively learned through well-designed OSL algorithms. Through extensive experiments on 11 state-of-the-art OSL algorithms, we identify two key strategies for improving one-shot FL: generating outliers and utilizing self-supervised tasks. Building on these findings, we introduce FedAdav, an adaptive algorithm that combines these OSL techniques to substantially enhance ensemble accuracy under label skew conditions. Our results demonstrate that combining multiple OSL signals can significantly improve federated learning performance, providing a strong foundation for future research aimed at addressing label skews in FL.

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

## A    MORE RELATED WORKS ON OSL

We follow a recent survey (Salehi et al., 2021) to introduce the OSL works.

In prior studies, OSL has three slightly different settings: anomaly detection (AD), open-set recognition (OSR), and out-of-distribution (OOD) detection (Salehi et al., 2021). Their goals are similar while the experimental settings are slightly different. AD regards one single class as known. OSR generally trains on a few classes and evaluates both the test accuracy of seen classes and the unknown detection capability between seen and unseen classes. OOD detection usually trains on a whole dataset and tests on other semantically different datasets. Under classification tasks, OSR and OOD detection become the same (Salehi et al., 2021). In summary, they share the same goal of distinguishing unknown samples from known samples. OSR and OOD detection have label supervision with an extra task of classifying seen classes. AD can be viewed as a special case of OSR and OOD detection where only one seen class exists.

For AD, (Zhou & Paffenroth, 2017) utilizes the reconstruction loss of auto-encoder trained on normal samples as the outlier score. ALOCC (Sabokrou et al., 2018) combines reconstruction loss with GAN loss to utilize the semantic-level similarity instead of pixel-level distance to distinguish outliers. Mem-AE (Gong et al., 2019) selects sparse latent features to store unique features for reconstructing normal samples. Old-is-Gold (Zaheer et al., 2020) utilizes the low-epoch generator to generate low-quality examples as outliers. DeepSVDD (Ruff et al., 2018) forces the latent features of auto-encoder to form a hyper-sphere and computes the outlier score by the distance to the center. GT (Golan & El-Yaniv, 2018) requires the model to predict which transformation is applied and regards the prediction error as outlier score. CSI (Tack et al., 2020) applies contrastive learning to distinguish shifted distribution from normal samples. Cutpaste (Li et al., 2021a) proposes cutting or scarring part of the image as outlier samples.

For OSR and OOD, OpenMax (Bendale & Boult, 2016) utilizes the logits before softmax to estimate the outlier score. G-OpenMax (Ge et al., 2017) trains GAN and interpolates embeddings between known classes to generate outlier samples. OSRCI (Neal et al., 2018) generates outliers on the embedding space with low probability of known classes. C2AE (Oza & Patel, 2019) uses auto-encoder reconstruction error of each class to identify outliers and applies extreme value theory (EVT) to set the threshold. CROSR (Yoshihashi et al., 2019a) proposes reconstructing the latent vector of each layer to preserve the information to distinguish known and unknown samples. GDFR (Perera et al., 2020a) integrates self-supervised learning with classification and inputs both the original and reconstructed image to extract richer features. PROSER (Zhou et al., 2021) regards the interpolation between known classes as outliers. OpenGAN (Kong & Ramanan, 2021) generates fake images between the embeddings of true samples and outlier examples. ODIN (Liang et al., 2017) discounts the logits during training and applies adversarial learning to pre-process the image to better distinguish outliers. In (Lee et al., 2018), softmax is replaced by the distance to the center of each class. Another work (Lee et al., 2017) trains GAN while maximizing the confidence entropy to generate outlier samples in the low confidence region. OE (Hendrycks et al., 2019a) trains with auxiliary outlier samples from other datasets to improve the capability of detecting outliers. (Hendrycks et al., 2019b) finds out that self-supervised learning tasks can improve outlier detection. (Yu & Aizawa, 2019) trains an ensemble of models and encourages diversity of outputs when training on outlier samples. G-ODIN (Hsu et al., 2020) trains separate headers for the outlier detection and closed-set classifier. (Serrà et al., 2019) observes that simpler OOD samples can lead to higher confidence and adds complexity measurement to the outlier score. Energy-based OOD (Liu et al., 2020) uses the energy function as the outlier score, which cares more about the maximum logit value. LogitNorm (Wei et al., 2022) normalizes the logits to address the over-confidence problem of deep neural networks. MOS (Huang & Li, 2021) divides the datasets into various semantic groups and trains an OSL model for each group. GradNorm (Huang et al., 2021) utilizes the gradient with respect to uniform probability distribution as the outlier score function. ReAct (Sun et al., 2021) limits the logits with an upper threshold to alleviate the overconfident predictions. BATS (Zhu et al., 2022) proposes truncated batch normalization to restrain the negative effects of extreme features. (Mirzaei et al., 2022) applies diffusion model with early stopping to generate high-quality outlier examples. A recent work (Tao et al., 2023) proposes to synthesize virtual outliers in latent space by sampling around the boundary of true samples.

# B  PROOFS

## B.1  NOTATIONS

Table 2 lists the notations used in our theoretical proof.

Table 2: Notations.

| Notation | Description |
|---|---|
| $\mathcal{X}$ | feature space |
| $\mathcal{Y}$ | label space |
| $\mathcal{D}$ | distribution space |
| $N$ | number of clients |
| $K$ | number of labels |
| $Y^i$ | label space of client $i$ |
| $D_i$ | distribution of client $i$ |
| $D_{[1:N]}$ | weighted distribution of all clients' local distributions |
| $D^\alpha$ | weighted distribution of $D$ and OOD set $O$ under weight $\alpha$ |
| $\pi_i$ | weight of $D_i$ in $D_{[1:N]}$ |
| $\mathcal{H}$ | hypothesis space |
| $\mathcal{L}$ | loss function |
| $\mathbf{R}$ | risk function, expectation of loss function |
| $S$ | set of samples |
| $f_c^i$ | closed-set classifier of client $i$, vector-valued function |
| $u^i$ | OOD detector of client $i$, real-valued function |
| $f_o^i$ | open-set classifier of client $i$, vector-valued function |
| $F$ | global ensemble function |
| $\varepsilon_n$ | error rate in correspondence to sample number $n$ |
| $m(D)$ | measure of a set $D$ |

## B.2  PROOF OF THEOREM 3.1

*Proof.* We notice that

$$
\mathbb{E}_{(x,y)\sim\mathcal{D}}\mathcal{L}(F(x);y) = \mathbb{E}_{(x,y)\sim\mathcal{D}}\left[-\log\frac{\sum_{i=1}^N u^i(x)f_c^i(x)_y}{\sum_{i=1}^N u^i(x)}\right]
$$

$$
= \mathbb{E}_{(x,y)\sim\mathcal{D}}\left[-\log\frac{\sum_{i=1}^N \mathbf{1}_{y\in Y^i}u^i(x)f_c^i(x)_y}{\sum_{i=1}^N u^i(x)}\right]
$$

$$
= \mathbb{E}_{(x,y)\sim\mathcal{D}}\left[-\log\frac{\sum_{i=1}^N \mathbf{1}_{y\in Y^i}u^i(x)f_c^i(x)_y}{\sum_{i=1}^N \mathbf{1}_{y\in Y^i}u^i(x)} - \log\frac{\sum_{i=1}^N \mathbf{1}_{y\in Y^i}u^i(x)}{\sum_{i=1}^N u^i(x)}\right]
$$

$$
\leq \mathbb{E}_{(x,y)\sim\mathcal{D}}\left[-\frac{\sum_{i=1}^N \mathbf{1}_{y\in Y^i}u^i(x)\log f_c^i(x)_y}{\sum_{i=1}^N \mathbf{1}_{y\in Y^i}u^i(x)} - \log\frac{\sum_{i=1}^N \mathbf{1}_{y\in Y^i}u^i(x)}{\sum_{i=1}^N u^i(x)}\right]
$$

$$
= \mathbb{E}_{(x,y)\sim\mathcal{D}}\left[\frac{\sum_{i=1}^N \mathbf{1}_{y\in Y^i}u^i(x)\mathcal{L}(f_c^i(x);y)}{\sum_{i=1}^N \mathbf{1}_{y\in Y^i}u^i(x)} + \mathcal{L}(u(x);y)\right]
$$

$$
\leq \varepsilon + \delta
$$

In the first step, if class $y$ is unseen in client $i$, its model should predict 0% for class $y$, i.e. $f_c^i(x)_y = 0$. If the loss of seen classes in each client $\mathbf{1}_{y\in Y^i}\mathcal{L}(f_c^i(x);y) \leq \varepsilon$, the first part (closed-set classification loss) can be bounded by $\varepsilon$. The second part is exactly the OOD detection loss bounded by $\delta$.

□

### B.3 PROOF OF THEOREM 3.4

*Proof.* For a set of $\{f_c^i\}_{i=1}^N$ satisfying Definition 3.3, suppose they also satisfy Definition 3.2. We denote the optimal global model in $\mathcal{H}$ as $F^*$.

**Part I.** We first show that there exists a constant $C > 0$ which is irrelevant to $n$, such that for all $n$, there exists $S$ and $F = F(S)$, such that

$$C\varepsilon_n > \mathcal{L}(F(x); y) - \mathcal{L}(F^*(x); y),$$

holds on a subset $P$ of $D_{[1:N]}$, whose measure $m(P)/m(D_{[1:N]}) > 1 - r/2$.

Otherwise, $\exists n, \forall C > 0, \forall S, \forall F : \mathcal{L}(F(x); y) - \mathcal{L}(F^*(x); y) \geq C\varepsilon_n$ holds for a subset $A \subset D_{[1:N]}$ with measure $m(A)/m(D_{[1:N]}) > r/2$. Therefore,

$$
\begin{aligned}
\mathbf{R}_{D_{[1:N]}}(F) - \mathbb{E}_{(x,y)\sim D_{[1:N]}}\mathcal{L}(F^*(x); y) &= \mathbb{E}_{(x,y)\sim D_{[1:N]}}\left[\mathcal{L}(F(x); y) - \mathcal{L}(F^*(x); y)\right] \\
&\geq \mathbb{E}_{(x,y)\sim D_{[1:N]}}\left[\mathbf{1}_{(x,y)\sim A}\left(\mathcal{L}(F(x); y) - \mathcal{L}(F^*(x); y)\right)\right] \\
&\geq \frac{m(A)}{m(D_{[1:N]})}C\varepsilon_n > \frac{r}{2}C\varepsilon_n = C'\varepsilon_n.
\end{aligned}
$$

Since we now assume it holds for any $n$-element set $S$, then we can further take expectation over $S$:

$$\mathbb{E}_{S\sim D_{[1:N]}}\mathbf{R}_{D_{[1:N]}}(F) \geq C'\varepsilon_n + \mathbb{E}_{(x,y)\sim D_{[1:N]}}\mathcal{L}(F^*(x); y).$$

This contradicts Definition 3.2 when $C$ becomes large enough to make $C' > 1$.

**Part II.** The above proposition states that such inequality holds on $P$ with measure $m(P)/m(D_{[1:N]}) > 1 - r/2$. Since we have assumed $m(\{(x,y) : y \text{ is not seen in all } Y^i\})/m(D_{[1:N]}) = r > 0$, we can assure that the two sets have overlap.

For element $(x, y)$ belonging to the overlap, we utilize the above result:

$$
\begin{aligned}
C\varepsilon_n > \mathcal{L}(F(x); y) - \mathcal{L}(F^*(x); y) &= -\log\frac{\sum_{i=1}^N f_c^i(x)_y}{N} - \mathcal{L}(F^*(x); y) \\
&\geq -\log\frac{N-m}{N} - \mathcal{L}(F^*(x); y).
\end{aligned}
$$

Here $m = \sum_{i:y\notin Y^i} 1 \geq 1$ is the number of clients where $y$ is unseen. Thus when $\lim_{n\to\infty}\varepsilon_n = 0$, we have $\mathcal{L}(F^*(x); y) = -\log\frac{N-m}{N}$. However, since $\mathcal{H}$ is big enough, the optimal $F^*$ should ensure $\mathcal{L}(F^*(x); y) \to 0$. This leads to a contradiction and shows that $F$ won't satisfy Definition 3.2.

$\square$

### B.4 PROOF OF THEOREM 3.7

**Lemma B.1.** *Let* $\{1, \cdots, N\} = A \cup B, A \cap B = \varnothing, x_i \in [0, 1], B \neq \varnothing$, *and* $\sum_{i\in B} x_i > 0$. *Then*

$$F(x_1, \cdots, x_N) = \left(1 + \frac{\sum_{i\in A} x_i}{\sum_{i\in B} x_i}\right)\prod_{i\in B} x_i \prod_{i\in A}(1 - x_i) \leq 1.$$

*Proof.* For $k \in B$:

$$\frac{\partial F}{\partial x_k} = \prod_{i\in B, i\neq k} x_i \prod_{i\in A}(1 - x_i)\left[\frac{\sum_{i\in A} x_i \sum_{i\in B, i\neq k} x_i}{\left(\sum_{i\in B} x_i\right)^2} + 1\right] > 0,$$

thus

$$
\begin{aligned}
F(x_1, \cdots, x_N) \leq F|_{x_k=1, \forall k\in B} &= \left(1 + \frac{\sum_{i\in A} x_i}{|B|}\right)\prod_{i\in A}(1 - x_i) \\
&\leq \left(1 + \sum_{i\in A} x_i\right)\prod_{i\in A}(1 - x_i) = G(x_i)_{i\in A}.
\end{aligned}
$$

Then for $k \in A$:

$$\frac{\partial G}{\partial x_k} = \prod_{i \in A, i \neq k} (1 - x_i) \left( -x_k - \sum_{i \in A} x_i \right) < 0,$$

thus

$$F(x_1, \cdots, x_N) \leq G(x_i)_{i \in A} \leq G|_{x_i = 0, \forall k \in A} = 1.$$

$\square$

*Proof of Theorem 3.7.* Use the result in the proof of Theorem 3.1 and Lemma B.1:

$$\mathcal{L}(F(x); y) \leq -\frac{\sum_{i=1}^N \mathbf{1}_{y \in Y^i} u^i(x) \log f_c^i(x)_y}{\sum_{i=1}^N \mathbf{1}_{y \in Y^i} u^i(x)} - \log \frac{\sum_{i=1}^N \mathbf{1}_{y \in Y^i} u^i(x)}{\sum_{i=1}^N u^i(x)}$$

$$= -\frac{\sum_{i=1}^N \mathbf{1}_{y \in Y^i} u^i(x) \log f_c^i(x)_y}{\sum_{i=1}^N \mathbf{1}_{y \in Y^i} u^i(x)} + \log \left( 1 + \frac{\sum_{i=1}^N \mathbf{1}_{y \notin Y^i} u^i(x)}{\sum_{i=1}^N \mathbf{1}_{y \in Y^i} u^i(x)} \right)$$

$$\leq \sum_{i=1}^N \left[ -\mathbf{1}_{y \in Y^i} \log f_c^i(x)_y \right] + \log \left( \frac{1}{\prod_{i:y \in Y^i} u^i(x) \prod_{i:y \notin Y^i} (1 - u^i(x))} \right)$$

$$= -\sum_{i=1}^N \left[ \mathbf{1}_{y \in Y^i} \log f_c^i(x)_y + \mathbf{1}_{y \in Y^i} \log u^i(x) + \mathbf{1}_{y \notin Y^i} \log(1 - u^i(x)) \right]$$

$$= \sum_{i=1}^N \mathcal{L}(f_o^i(x); y).$$

Let $\alpha_i' = 1 - \frac{\pi_i}{\pi_{[1:N]}} (1 - \alpha_i) \in (0, 1), O_i' = \frac{\pi_i \alpha_i O_i}{\alpha_i' \pi_{[1:N]}} + \sum_{k \neq i} \frac{\pi_k D_k^{\alpha_k}}{\pi_{[1:N]} \alpha_i'}$, then $D_i^{\alpha_i'} = (1 - \alpha_i') D_i + \alpha_i' O_i' = D_{[1:N]}^{\alpha_{[1:N]}}$. Take expectation on the above inequality over $D_{[1:N]}^{\alpha_{[1:N]}}$:

$$\mathbf{R}_{D_{[1:N]}^{\alpha_{[1:N]}}}(F) = \mathbb{E}_{(x,y) \sim D_{[1:N]}^{\alpha_{[1:N]}}} \mathcal{L}(F(x); y) \leq \mathbb{E}_{(x,y) \sim D_{[1:N]}^{\alpha_{[1:N]}}} \sum_{i=1}^N \mathcal{L}(f_o^i(x); y)$$

$$= \sum_{i=1}^N \mathbb{E}_{(x,y) \sim D_i^{\alpha_i'}} \mathcal{L}(f_o^i(x); y) = \sum_{i=1}^N \mathbf{R}_{D_i^{\alpha_i'}}(f_o^i).$$

Suppose we have found $\{f_o^i\}_{i=1}^N$ satisfying Definition 3.6. Since $\inf_{G \in \mathcal{H}} \mathbf{R}_{D_{[1:N]}^{\alpha_{[1:N]}}}(G) = 0$, naturally $\inf_{g_i \in \mathcal{H}} \mathbf{R}_{D_i^{\alpha_i'}}(g_i) = 0$. Thus the loss of each local open-set classifier $\max_{i=1,\cdots,N} \mathbf{R}_{D_i^{\alpha_i'}}(f_o^i) < \varepsilon_n$. Then the loss of open-set ensemble $\mathbf{R}_{D_{[1:N]}^{\alpha_{[1:N]}}}(F) < N\varepsilon_n$, and goes to 0 as the number of sample $n \to \infty$, which satisfies Definition 3.5.

$\square$

## C FURTHER EXPERIMENTS

This section shows more empirical results of OSL algorithms on FL settings.

### C.1 DETAILED SETUPS

The statistics of the datasets used in our experiments are summarized in 3. For MNIST, in accordance with OpenOOD code framework, we convert the images to 3 channels (RGB) before inputting the LeNet. For CIFAR-10 and CIFAR-100, ResNet-18 has batch normalization layers, which harms the effectiveness of FedOV. According to the solutions in FedOV framework, we only keep the data destruction loss and combine real samples and destructed samples into a batch. For algorithms with hyper-parameters, we use the default values in OpenOOD. Experiments are conducted on a single NVIDIA GeForce RTX 3090 GPU.

Table 3: Basic information of datasets we use.

| Datasets | Train set size | Test set size | Data dimension | #classes |
|---|---|---|---|---|
| MNIST | 60,000 | 10,000 | 784 | 10 |
| CIFAR-10 | 50,000 | 10,000 | 3,072 | 10 |
| CIFAR-100 | 50,000 | 10,000 | 3,072 | 100 |

## C.2 SUPPLEMENTARY EXPERIMENTAL RESULTS

In addition to the primary experiments, we evaluated four additional baselines—CE+GEN, CE+MLS, RegMixup+MSP, and CE+ReAct—with their performance results summarized in Table 4. These methods, which apply standard closed-set training techniques combined with simple OSL strategies, did not achieve top-2 accuracy in any of the one-shot FL configurations tested. Although these methods achieve SOTA AUROC in OpenOOD benchmark, they do not perform well in one-shot FL setups.

The results demonstrate that models relying solely on traditional classification approaches or isolated OSL methods struggle to generalize in open-set scenarios, where detecting outliers and handling imbalanced data distributions are crucial. These findings underscore the importance of combining multiple OSL signals, as implemented in FedAdav, which adaptively integrates outlier detection and self-supervised learning for significantly improved ensemble performance in one-shot FL.

Table 4: Comparing with 4 additional baseline OSL algorithms on one-shot FL label skews. All experiments are repeated with three different seeds.

| Dataset | Partition | FedAdav | CE+GEN | CE+MLS | RegMixup+MSP | CE+ReAct |
|---|---|---|---|---|---|---|
| MNIST | $\#C = 1$ | $\mathbf{70.9\%}_{\pm 0.8\%}$ | $18.2\%_{\pm 0.4\%}$ | $18.2\%_{\pm 0.4\%}$ | $13.8\%_{\pm 0.4\%}$ | $18.2\%_{\pm 0.4\%}$ |
| | $\#C = 2$ | $\mathbf{78.3\%}_{\pm 10.5\%}$ | $61.6\%_{\pm 2.4\%}$ | $61.7\%_{\pm 2.2\%}$ | $57.4\%_{\pm 4.5\%}$ | $60.9\%_{\pm 2.5\%}$ |
| | $\#C = 3$ | $\mathbf{78.1\%}_{\pm 5.8\%}$ | $71.1\%_{\pm 5.5\%}$ | $71.0\%_{\pm 5.5\%}$ | $69.0\%_{\pm 6.8\%}$ | $71.5\%_{\pm 4.9\%}$ |
| | $p_k \sim Dir(0.1)$ | $\mathbf{80.9\%}_{\pm 1.8\%}$ | $79.5\%_{\pm 3.4\%}$ | $79.5\%_{\pm 3.3\%}$ | $73.4\%_{\pm 5.2\%}$ | $78.3\%_{\pm 2.6\%}$ |
| CIFAR-10 | $\#C = 1$ | $\mathbf{51.7\%}_{\pm 0.7\%}$ | $8.4\%_{\pm 0.1\%}$ | $8.4\%_{\pm 0.1\%}$ | $7.3\%_{\pm 0.1\%}$ | $8.5\%_{\pm 0.2\%}$ |
| | $\#C = 2$ | $\mathbf{58.4\%}_{\pm 5.5\%}$ | $44.9\%_{\pm 1.2\%}$ | $44.7\%_{\pm 1.3\%}$ | $46.8\%_{\pm 0.3\%}$ | $45.3\%_{\pm 1.2\%}$ |
| | $\#C = 3$ | $\mathbf{61.4\%}_{\pm 4.4\%}$ | $57.9\%_{\pm 4.4\%}$ | $57.4\%_{\pm 4.6\%}$ | $58.6\%_{\pm 4.7\%}$ | $57.9\%_{\pm 4.1\%}$ |
| | $p_k \sim Dir(0.1)$ | $\mathbf{70.0\%}_{\pm 2.0\%}$ | $63.4\%_{\pm 3.2\%}$ | $63.0\%_{\pm 3.3\%}$ | $65.2\%_{\pm 3.1\%}$ | $62.7\%_{\pm 4.0\%}$ |
| CIFAR-100 | $\#C = 1$ | $\mathbf{13.9\%}_{\pm 0.6\%}$ | $0.5\%_{\pm 0.0\%}$ | $0.5\%_{\pm 0.0\%}$ | $0.5\%_{\pm 0.0\%}$ | $0.5\%_{\pm 0.0\%}$ |
| | $\#C = 2$ | $\mathbf{15.1\%}_{\pm 1.5\%}$ | $8.7\%_{\pm 0.5\%}$ | $8.7\%_{\pm 0.5\%}$ | $9.0\%_{\pm 0.7\%}$ | $9.2\%_{\pm 0.4\%}$ |
| | $\#C = 3$ | $\mathbf{20.9\%}_{\pm 0.8\%}$ | $13.0\%_{\pm 1.3\%}$ | $13.1\%_{\pm 1.3\%}$ | $15.6\%_{\pm 1.0\%}$ | $13.7\%_{\pm 1.0\%}$ |
| | $p_k \sim Dir(0.1)$ | $\mathbf{31.2\%}_{\pm 0.5\%}$ | $26.7\%_{\pm 0.1\%}$ | $26.7\%_{\pm 0.2\%}$ | $28.0\%_{\pm 0.3\%}$ | $26.6\%_{\pm 0.2\%}$ |

Table 5: Test accuracy of OSL algorithms on one-shot FL label skews. We apply 95% TPR threshold to algorithms in OpenOOD for voting according to the convention. All experiments are repeated with three different seeds. In each line, the highest accuracy is bold and the second highest is underlined.

| Dataset | Partition | FedAdav | FedOV | CE+MSP (closed-set) | RotPred +PixMix | LogitNorm+ PixMix+MSP | PixMix +MSP | RotPred | LogitNorm +MSP |
|---|---|---|---|---|---|---|---|---|---|
| MNIST | $\#C = 3$ | $\mathbf{78.1\%}_{\pm 5.8\%}$ | $72.3\%_{\pm 3.8\%}$ | $70.6\%_{\pm 5.4\%}$ | $\underline{75.4\%}_{\pm 2.7\%}$ | $72.4\%_{\pm 1.8\%}$ | $70.5\%_{\pm 5.4\%}$ | $75.2\%_{\pm 3.0\%}$ | $72.7\%_{\pm 2.5\%}$ |
| | $p_k \sim Dir(0.1)$ | $80.9\%_{\pm 1.8\%}$ | $82.4\%_{\pm 7.7\%}$ | $79.1\%_{\pm 3.1\%}$ | $82.0\%_{\pm 2.4\%}$ | $\mathbf{83.5\%}_{\pm 3.9\%}$ | $79.2\%_{\pm 3.2\%}$ | $81.9\%_{\pm 2.6\%}$ | $\underline{83.4\%}_{\pm 4.3\%}$ |
| CIFAR-10 | $\#C = 3$ | $\mathbf{61.4\%}_{\pm 4.4\%}$ | $54.4\%_{\pm 5.3\%}$ | $58.8\%_{\pm 3.5\%}$ | $\underline{65.1\%}_{\pm 3.2\%}$ | $57.2\%_{\pm 4.5\%}$ | $58.9\%_{\pm 3.7\%}$ | $65.2\%_{\pm 3.2\%}$ | $57.1\%_{\pm 4.4\%}$ |
| | $p_k \sim Dir(0.1)$ | $\underline{70.0\%}_{\pm 2.0\%}$ | $\mathbf{73.1\%}_{\pm 7.7\%}$ | $63.3\%_{\pm 2.3\%}$ | $69.3\%_{\pm 3.6\%}$ | $61.5\%_{\pm 3.6\%}$ | $63.3\%_{\pm 2.6\%}$ | $69.4\%_{\pm 3.8\%}$ | $61.5\%_{\pm 3.5\%}$ |
| CIFAR-100 | $\#C = 3$ | $\mathbf{20.9\%}_{\pm 0.8\%}$ | $\underline{19.7\%}_{\pm 0.9\%}$ | $14.9\%_{\pm 0.8\%}$ | $16.7\%_{\pm 0.8\%}$ | $14.6\%_{\pm 0.9\%}$ | $15.1\%_{\pm 0.5\%}$ | $16.0\%_{\pm 0.2\%}$ | $14.6\%_{\pm 0.7\%}$ |
| | $p_k \sim Dir(0.1)$ | $\mathbf{31.2\%}_{\pm 0.5\%}$ | $\underline{30.4\%}_{\pm 0.5\%}$ | $26.8\%_{\pm 0.3\%}$ | $28.8\%_{\pm 0.3\%}$ | $24.9\%_{\pm 0.6\%}$ | $26.8\%_{\pm 0.0\%}$ | $28.5\%_{\pm 0.2\%}$ | $25.0\%_{\pm 0.5\%}$ |

We also put results on slightly label skew setups in Table 5. When label skews are slight, the accuracy degradation from IID settings is small. In such cases, FedAdav achieves similar accuracy with FedOV and RotPred.

## C.3 MORE RESULTS ON THRESHOLDS

FedOV and MSP-based post-processor have explicit confidence probability. In Table 6, we compare the explicit confidence with 95% TPR threshold. As we can see, FedOV works better with the explicit confidence probability, while the other five methods with MSP-based post-processor achieve higher accuracy with 95% TPR threshold.

Table 6: Comparing natural threshold with 95% TPR threshold. In each cell, the first accuracy uses the explicit confidence probability while the second accuracy uses 95% TPR threshold. The higher accuracy in each cell is underlined. All experiments are repeated with three different random seeds.

| Dataset | Partition | FedOV | CE+MSP (closed-set) | LogitNorm+ PixMix+MSP | PixMix +MSP | LogitNorm +MSP | RegMixup +MSP |
|---|---|---|---|---|---|---|---|
| MNIST | #C = 1 | 68.2%±1.9%/64.7%±2.2% | 17.7%±0.5% / 18.3%±0.4% | 10.3%±0.0% / 13.2%±0.3% | 17.7%±0.5% / 18.3%±0.4% | 10.3%±0.0% / 13.2%±0.3% | 13.7%±0.4% / 13.8%±0.4% |
| | #C = 2 | 60.0%±7.8%/72.5%±11.2% | 48.5%±7.3%/60.1%±3.7% | 41.9%±4.0%/54.4%±0.8% | 48.6%±7.3%/60.1%±3.7% | 41.7%±3.6%/54.1%±0.5% | 47.9%±4.2%/57.4%±4.5% |
| | #C = 3 | 72.3%±3.8%/84.9%±2.7% | 60.0%±7.5%/70.6%±3.4% | 61.0%±4.1%/72.4%±1.8% | 59.9%±7.6%/70.5%±5.4% | 60.8%±4.1%/72.7%±2.5% | 62.7%±7.2%/69.0%±6.8% |
| | $p_k \sim Dir(0.1)$ | 82.4%±7.7%/88.8%±3.7% | 72.5%±7.5%/79.1%±3.1% | 75.7%±9.0%/83.5%±3.9% | 78.4%±12.5%/79.2%±3.2% | 82.9%±3.7%/83.4%±4.3% | 72.0%±3.8%/73.4%±5.2% |
| CIFAR-10 | #C = 1 | 26.2%±4.1%/28.5%±3.6% | 8.2%±0.2%/8.4%±0.1% | 10.0%±0.0%/7.5%±0.4% | 8.2%±0.1%/8.5%±0.1% | 10.0%±0.0%/7.6%±0.4% | 6.9%±0.2%/7.3%±0.1% |
| | #C = 2 | 45.4%±2.5%/50.8%±3.3% | 41.1%±1.9%/45.4%±1.1% | 37.7%±2.9%/41.4%±3.3% | 41.0%±1.6%/45.2%±1.0% | 37.8%±3.0%/41.2%±3.5% | 42.2%±2.0%/46.8%±0.3% |
| | #C = 3 | 54.4%±5.3%/61.0%±1.5% | 52.3%±6.1%/58.8%±3.5% | 51.7%±5.7%/57.2%±4.5% | 52.1%±5.9%/58.9%±3.7% | 51.6%±5.6%/57.1%±4.4% | 53.9%±6.0%/58.6%±4.7% |
| | $p_k \sim Dir(0.1)$ | 73.1%±7.7%/71.5%±3.1% | 60.6%±4.5%/63.3%±2.3% | 59.9%±4.2%/61.5%±3.6% | 60.7%±4.9%/63.3%±2.6% | 60.0%±4.4%/61.5%±3.5% | 63.8%±4.4%/65.2%±3.1% |
| CIFAR-100 | #C = 1 | 8.9%±0.6%/7.9%±0.9% | 0.5%±0.0%/0.5%±0.0% | 0.4%±0.0%/0.4%±0.1% | 0.5%±0.0%/0.5%±0.0% | 0.4%±0.1%/0.4%±0.0% | 0.5%±0.0%/0.5%±0.0% |
| | #C = 2 | 15.7%±0.5%/13.2%±1.1% | 6.3%±0.8%/9.2%±0.4% | 5.9%±0.8%/8.9%±0.7% | 6.4%±1.1%/9.3%±0.5% | 6.3%±0.8%/9.0%±0.8% | 6.7%±0.9%/9.0%±0.7% |
| | #C = 3 | 19.7%±0.9%/18.4%±1.1% | 9.9%±1.4%/14.9%±0.8% | 10.6%±1.1%/14.6%±0.9% | 10.1%±1.4%/15.1%±0.5% | 10.5%±1.3%/14.6%±0.7% | 11.8%±1.3%/15.6%±1.0% |
| | $p_k \sim Dir(0.1)$ | 30.4%±0.5%/33.1%±0.4% | 26.2%±0.2%/26.8%±0.3% | 25.1%±0.8%/24.9%±0.6% | 25.8%±1.1%/26.8%±0.0% | 25.0%±0.6%/25.0%±0.5% | 25.9%±1.7%/28.0%±0.3% |

We also vary the TPR threshold values on MNIST and CIFAR-100. Results are shown in Figure 2. FedOV is good when TPR is over 90%, while RotPred works better at about 80%. Thresholds can also be set by methods like extreme value theory (EVT). However, the effects are similar to TPR where the threshold should include the majority of training data as in-distribution data and exclude the minority as OOD data. We show the results of RotPred in Table 7. As we can see, tuning threshold with mean and standard deviation achieves quite similar accuracy with tuning the TPR threshold value. Thus, we focus on TPR threshold in this paper.

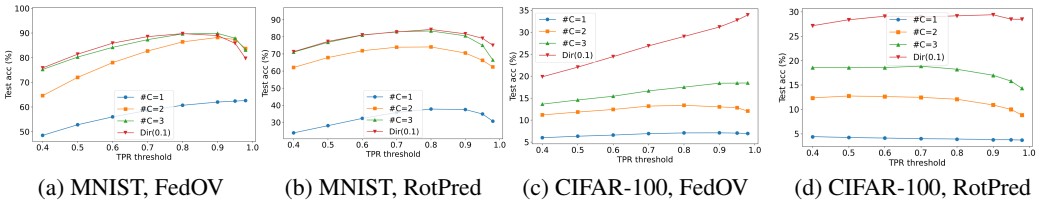

| (a) MNIST, FedOV | (b) MNIST, RotPred | (c) CIFAR-100, FedOV | (d) CIFAR-100, RotPred |
|---|---|---|---|

Figure 2: Test accuracy on MNIST and CIFAR-100 using different TPR threshold values among $\{0.4, 0.5, 0.6, 0.7, 0.8, 0.9, 0.95, 0.98\}$.

Table 7: Comparing different threshold methods. TPR threshold values are among $\{0.4, 0.5, 0.6, 0.7, 0.8, 0.9, 0.95, 0.98\}$. The extreme value threshold is tested among mean plus $\{1, 2, 3, 4\}$ standard deviations.

| Dataset | Partition | RotPred (TPR $0.4 \sim 0.98$) | RotPred ($\mu + 1 \sim 4 * \sigma$) |
|---|---|---|---|
| MNIST | $\#C = 1$ | $23.9\% \sim 37.8\%$ | $21.3\% \sim \mathbf{38.4\%}$ |
| | $\#C = 2$ | $62.1\% \sim \mathbf{74.1\%}$ | $56.6\% \sim 73.2\%$ |
| | $\#C = 3$ | $66.5\% \sim \mathbf{83.4\%}$ | $58.6\% \sim 82.3\%$ |
| | $p_k \sim Dir(0.1)$ | $71.3\% \sim \mathbf{84.3\%}$ | $63.2\% \sim 84.1\%$ |
| CIFAR-10 | $\#C = 1$ | $45.0\% \sim \mathbf{48.7\%}$ | $44.7\% \sim 48.5\%$ |
| | $\#C = 2$ | $54.2\% \sim \mathbf{59.3\%}$ | $50.0\% \sim 58.8\%$ |
| | $\#C = 3$ | $60.6\% \sim \mathbf{65.9\%}$ | $54.0\% \sim 65.8\%$ |
| | $p_k \sim Dir(0.1)$ | $61.7\% \sim \mathbf{72.3\%}$ | $69.9\% \sim 72.0\%$ |
| CIFAR-100 | $\#C = 1$ | $3.7\% \sim \mathbf{4.4\%}$ | $3.5\% \sim 3.9\%$ |
| | $\#C = 2$ | $8.9\% \sim \mathbf{12.8\%}$ | $6.4\% \sim 11.5\%$ |
| | $\#C = 3$ | $14.3\% \sim \mathbf{18.9\%}$ | $10.4\% \sim 17.5\%$ |
| | $p_k \sim Dir(0.1)$ | $27.1\% \sim \mathbf{29.4\%}$ | $27.8\% \sim 29.2\%$ |

## C.4 EXPERIMENTS ON CENTRALIZED OSL SETTINGS

In this section, we evaluate the performance aspects of limited seen classes in one-shot FL ensemble, which are seldom studied in prior OSL works. Specifically, we conduct experiments to compare OSL algorithm performance on $k = 1, 2, 4, 8$ seen classes. We conduct the centralized OSL experimental settings on CIFAR-10, training 200 epochs on the first $k$ classes and testing on all 10 classes. AUROC results on outlier scores are reported in Figure 3. As we can see, except FedAdav, FedOV and

RotPred, other SOTA OSL algorithms do not work when there is only one seen class. FedAdav and FedOV have similar AUROC, since RotPred training is disabled for certain classes.

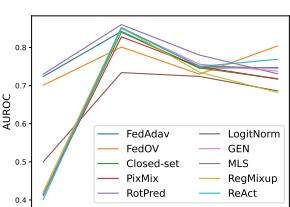

Figure 3: AUROC in centralized training on CIFAR-10 with varying number of seen classes.

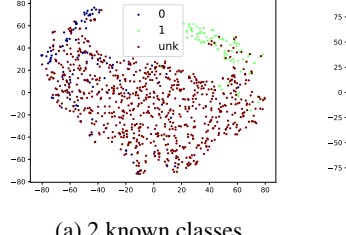

(a) 2 known classes

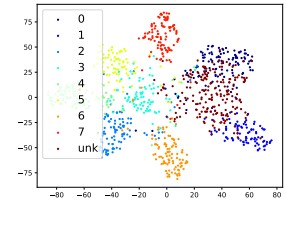

(b) 8 known classes

Figure 4: T-SNE visualization on extracted features of CIFAR-10.

We also find that the metrics of AUROC and one-shot FL ensemble accuracy are not always consistent. In terms of AUROC, post-processing algorithms (GEN, MLS and ReAct) have advantage against closed-set training with MSP, but they achieve similar accuracy on one-shot FL ensemble tests.

To explore the learned features of different OSL algorithms, we conduct t-SNE visualization on the features of the last layer. As shown in Figure 4, when there are only 2 known classes, the representation of known class can be highly mixed with unknown classes. Such challenge is under explored in prior OSL works and deserves further studies.

## C.5 T-SNE Visualization

In this section, we conduct t-SNE visualization of the last-layer features learned by different OSL algorithms. We train 200 epochs on the first $k = 1, 2, 4, 8$ classes on CIFAR-10, and the other $(10 - k)$ classes are regarded as unknown classes during the test.

Comparing among each row in Figure 5, the representation learned by FedOV and RotPred can better separate the known and unknown classes. The representation of unknown classes (brownish red) have smaller overlap with known classes in FedOV and RotPred, compared with the other four algorithms. The visualization verifies the effectiveness of the learned features by FedOV and RotPred. Thus, incorporating generated outliers or self-supervised tasks is a promising direction to address the OSL problem, especially with limited known classes.

Comparing among each column in Figure 5, we can observe that the fewer number of known classes, the more difficult OSL algorithms learn a good representation to differentiate unknown classes. For example, in the first row where there is only one known class, the representation of known and unknown classes are mixed. This illustrates the under-explored challenge of limited known classes in prior OSL studies and one-shot FL ensemble.

## C.6 Running Time

We record the average running time per epoch of different OSL algorithms in Table 8. The experiments are conducted on a single 3090 GPU. As we can see, compared with closed-set training baseline, FedAdav and FedOV take about 40% more training time, due to the computation of generated outliers. RotPred takes few extra training time when there are not too many classes ($\#C = 1, 2, 3$), while taking an extra 20% to 50% time on Dirichlet-based label skews where there are more classes. When there are many classes, the task of predicting correct class while predicting correct rotation becomes increasingly complicated, which may lead to higher computation costs. These two methods achieve better one-shot federated ensemble accuracy at the cost of more computations.

PixMix directly conducts data augmentation, which has similar computation time compared to closed-set learning. RegMixup and LogitNorm further speed up the training of closed-set learning, probably due to their regularization. LogitNorm normalizes the logits to a smaller scale than closed-set training,

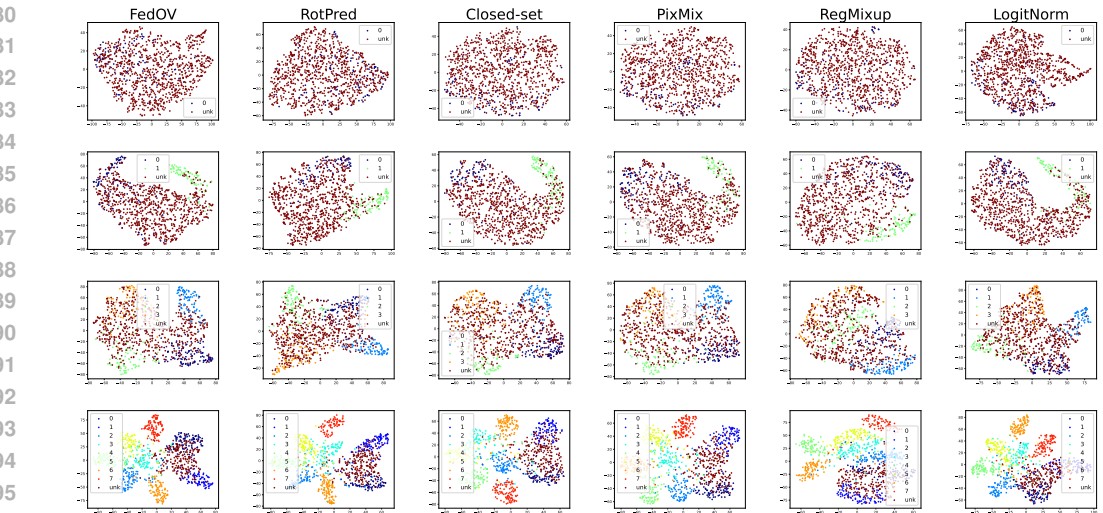

Figure 5: T-SNE visualization of the feature learned by OSL algorithms on CIFAR-10. The four rows are trained with 1, 2, 4, 8 known classes respectively. Other classes are marked as "unk" (unknown).

which may speed up the back-propagation process. However, these three methods do not achieve good FL ensemble accuracy.

Table 8: Comparing running time of different OSL methods. We record the average training time (in seconds) per epoch on the first client of CIFAR-10 dataset.

| Partition | FedAdav | Closed-set | FedOV | RotPred | PixMix | RegMixup | LogitNorm |
|---|---|---|---|---|---|---|---|
| $\#C = 1$ | 5.9 | 3.9 | 5.7 | 3.9 | 4.1 | 3.0 | 2.4 |
| $\#C = 2$ | 5.3 | 3.3 | 4.9 | 3.5 | 3.5 | 2.7 | 2.2 |
| $\#C = 3$ | 8.0 | 5.5 | 7.7 | 5.6 | 5.8 | 4.0 | 3.0 |
| $p_k \sim Dir(0.1)$ | 8.6 | 5.7 | 7.8 | 8.3 | 5.8 | 4.0 | 3.0 |

