# OpenReview forum: "Open-Set Learning for Addressing Label Skews in One-Shot Federated Learning"
_ICLR.cc/2025/Conference — Submitted to ICLR 2025_

### Official Review · Reviewer_87KU · 2024-11-01

**Soundness:** 3
**Presentation:** 3
**Contribution:** 2
**Rating:** 5
**Confidence:** 3

**Summary:**

This paper tackles the issue of label skews in one-shot federated learning (FL) by integrating open-set learning (OSL) techniques. The authors provide a theoretical analysis proving the learnability of one-shot FL ensembles with OSL algorithms and propose FedAdav, an adaptive algorithm that combines multiple OSL signals to improve ensemble accuracy under label skews. Extensive experiments demonstrate that FedAdav outperforms state-of-the-art OSL algorithms in severe label skew conditions.

**Strengths:**

- This paper theoretically proves that closed-set learning cannot effectively address label skews, whereas integrating OSL into FL could ensure the learnability of one-shot federated learning, which is the main contribution and a significant exploration.
- The theoretical analysis and empirical results are of reasonable quality, providing support for the proposed method.
- The paper is generally well-structured, but some sections could benefit from more detailed explanations.

**Weaknesses:**

The paper makes a reasonable contribution to the field of federated learning, yet there are areas that could be improved.
-  Although the theoretical part seems solid, the proposed algorithm appears somewhat incremental, which shares similarities with existing OSL methods. I suggest the authors to explore the algorithm's performance in other federated learning challenges such as data heterogeneity with different Dirichlet distributions to demonstrate its broader application potential.
- The empirical results are somewhat limited and could be strengthened by additional experiments on more diverse datasets or real-world applications. I recommend the authors conduct experiments in practical application areas such as natural language processing to validate the algorithm's effectiveness in different domains. Additionally, consider testing on more challenging datasets, such as large-scale image datasets and multilingual text datasets, to demonstrate the algorithm's robustness and generalization capabilities.
- The presentation of the theoretical proofs could be made more accessible to readers who are not experts in the field. I suggest the authors add more intuitive explanations at key steps, such as using charts to illustrate the proposed algorithm.

**Questions:**

Could the authors provide further insights into the scenarios where FedAdav might underperform compared to other methods? The paper could benefit from a more detailed discussion on the limitations of the proposed method, including scenarios where FedAdav might not perform as expected.

---

### Official Review · Reviewer_WbJn · 2024-11-02

**Soundness:** 3
**Presentation:** 3
**Contribution:** 3
**Rating:** 5
**Confidence:** 4

**Summary:**

The paper addresses the challenge of label skew in one-shot federated learning (FL), where clients communicate with the server only once, and class distributions across clients are imbalanced. Existing open-set learning (OSL) methods, like FedOV, help by identifying unknown samples but are limited in flexibility. The paper proposes an adaptive algorithm, FedAdav, combining multiple OSL signals to improve accuracy under label skew. The theoretical contribution proves the learnability of one-shot FL with OSL, and extensive experiments show FedAdav's effectiveness in enhancing performance over other state-of-the-art (SOTA) OSL methods.

**Strengths:**

1. The paper introduces a novel approach, FedAdav, that combines multiple OSL methods, enhancing the handling of label skew in one-shot FL. This combination of OSL signals is innovative for managing highly imbalanced class distributions across federated clients.

2. The theoretical analysis is rigorous, with proofs supporting the benefits of using OSL for addressing label skew in one-shot FL.

3. The paper is well-organized, presenting both theoretical foundations and empirical results to support the proposed approach. The steps in the algorithm and experiment setups are clearly described.

**Weaknesses:**

1. My biggest concern is about the setting of this paper, which only considers the one-shot FL. In a more general case, we usually have multiple communication rounds and the model will be aggregated in each round. However, the paper only focuses on one-shot FL, where clients communicate only once and the models are not aggregated. Instead, only the model predictions are aggregated. This limitation may restrict the effectiveness (at least from the theoretical part) in the general multi-round communication settings.

2. Limited Real-World Data Testing: While the experiments use standard datasets like MNIST and CIFAR, these do not fully represent the complexities of label skew in real-world FL applications, such as varying regional disease prevalence in healthcare. Adding real-world data (if any) could strengthen the practical relevance of FedAdav.

**Questions:**

Please carefully address W1.

---

### Official Review · Reviewer_w5yT · 2024-11-03

**Soundness:** 2
**Presentation:** 2
**Contribution:** 2
**Rating:** 5
**Confidence:** 3

**Summary:**

This paper explores the effectiveness of open-set learning (OSL) in improving one-shot federated learning (FL), especially when facing label skews. The authors provide a theoretical proof of OSL's benefits and propose a new method, FedAdav, combining features from previous works.

**Strengths:**

The paper includes a comprehensive theoretical analysis proving the utility of OSL in the context of one-shot FL.
Extensive experimental comparisons with multiple baselines are provided.

**Weaknesses:**

The proposed method lacks innovation, as it is simply a combination of the loss functions from two existing works (FedOV and RotPred).
The experiments use simple models and datasets, making it difficult to effectively validate the proposed method's robustness and general applicability.

**Questions:**

1.FedAdav simply combines the loss functions from FedOV and RotPred. What is the purpose of doing so? How does this enhance the method’s capability?
2.The trade-off parameters in FedAdav are derived through experimental results. How can you ensure that the same parameter settings would work effectively if the dataset or model changes?
3. The experiments only use very simple CNN models, such as LeNet, on simple datasets. Can such a simple setup properly evaluate the impact of OSL in one-shot FL? Why did the authors not use more sophisticated models, such as Transformer-based models, on more complex datasets?

---

### Official Review · Reviewer_man1 · 2024-11-03

**Soundness:** 2
**Presentation:** 3
**Contribution:** 2
**Rating:** 3
**Confidence:** 4

**Summary:**

This paper studied open-set learning (OSL) for one-shot federated learning under label shifts. It proved the learnability of one-shot federated learning ensembles with open-set learning. Then it introduced a FedAdav algorithm to combine multiple OSL approaches. Experimental results showed that FedAdav could outperform SOTA baselines.

**Strengths:**

**Originality:** It proved the learnability of one-shot FL ensembles with OSL. Specifically, this theoretical analysis highlighted the importance of OOD detection. By combining multiple OSL methods, the proposed FedAdav algorithm achieved superior performance than baselines.

**Quality:** Theorem 3.7 showed the impact of OOD detection function in proving the learnability of one-shot FL ensembles. Experimental results confirmed the effectiveness of the combination of multiple OSL methods in one-shot FL problems.

**Clarity:** The paper was well-written. The motivation of the proposed theory and algorithm is clearly illustrated.

**Significance:** It provides theoretical supports for analyzing one-shot FL ensembles with OSL.

**Weaknesses:**

(1) The connection between the theoretical analysis and the proposed FedAdav algorithm is weak. Theorem 3.7 shows that OSL is key to improving the performance of one-shot FL. However, it generally works well for all OSL methods, including baselines FedOV and RotPred. It is unclear what properties a OOD detection approach require to enhance one-shot FL.

(2) The proposed FedAdav is simple combination of FedOV and RotPred. As illustrated in section 4.1, both approaches have some limitations. It is not explained why the simple combination of FedOV and RotPred can solve these limitations. Moreover, it is confusing why the combination of FedOV and RotPred enables better learnability of one-shot FL ensembles in Theorem 3.7.

(3) The parameter sensitivity of FedAdav is not analyzed. There are two key hyperparameters: $T_{check}$ and $\tau$. Both affects how FedOV and RotPred loss function are performed in the proposed algorithm.

**Questions:**

(1) The server updating procedures of Algorithm 1 can be further explained. Will the prediction on every testing example be executed on the server? Is the the known confidence of the input $u^i(x)$ required to provide the ensemble prediction?

(2) What are the implications of $\alpha$ in Definition 3.5 and Definition 3.6? How will it affect the design of practical algorithm, e.g., FedAdav without $\alpha$?

(3) In Table 1, FedAdav has very large standard deviation on MNIST (#C=2). This phenomenon can be further explained.

---

### Meta-Review · Area_Chair_V8Vs · 2024-12-22

**Metareview:**

**Summary:** This paper tackles the challenge of label skews in one-shot federated learning (FL) by integrating open-set learning (OSL) techniques. The authors provide theoretical analysis proving the learnability of one-shot FL ensembles using OSL methods and propose FedAdav, an adaptive algorithm that combines OSL signals from existing methods like FedOV and RotPred.

**Decision:** Although the paper tackles an interesting problem, it falls short on several critical aspects, as noted by the reviewers. First, the proposed algorithm, FedAdav, lacks novelty as it is essentially a combination of existing methods (FedOV and RotPred) without sufficient justification for its effectiveness in addressing the limitations of these methods. Second, the theoretical analysis, while rigorous, does not connect strongly to the proposed algorithm, leaving the practical implications unclear. Third, the empirical evaluation is limited to simple datasets and models, failing to demonstrate the robustness or generalizability of the method to real-world FL scenarios or more challenging datasets. Finally, the focus on one-shot FL restricts the broader applicability of the work, as most practical FL scenarios involve multiple communication rounds. These limitations collectively outweigh the contributions, leading to a unanimous decision to reject.

**Additional Comments On Reviewer Discussion:**

The authors did not respond to the reviewers' comments during the rebuttal period.

---

### Decision · Program_Chairs · 2025-01-22

Reject